# The color phi phenomenon: Not so special, after all?

**Lars Keuninckx** [ID]☯¤*, **Axel Cleeremans**☯

Consciousness, Cognition & Computation Group (CO3), Center for Research in Cognition & Neurosciences (CRCN), Université Libre de Bruxelles, Bruxelles, Belgium

☯ These authors contributed equally to this work.
¤ Current address: Interuniversity Microelectronics Centre (IMEC), Leuven, Belgium
* lars.keuninckx@gmail.com

**Data Availability Statement:** All data underlying our results stem from model simulations in the form of Python scripts and are included in the Supporting Material.

**Funding:** AC and LK were supported by the European Research Council under the Advanced

## Abstract

We show how anomalous time reversal of stimuli and their associated responses can exist in very small connectionist models. These networks are built from dynamical toy model neurons which adhere to a minimal set of biologically plausible properties. The appearance of a "ghost" response, temporally and spatially located in between responses caused by actual stimuli, as in the phi phenomenon, is demonstrated in a similar small network, where it is caused by priming and long-distance feedforward paths. We then demonstrate that the color phi phenomenon can be present in an echo state network, a recurrent neural network, without explicitly training for the presence of the effect, such that it emerges as an artifact of the dynamical processing. Our results suggest that the color phi phenomenon might simply be a feature of the inherent dynamical and nonlinear sensory processing in the brain and in and of itself is not related to consciousness.

## Author summary

Well known visual illusions are used to investigate how objective physical stimuli translate to subjective conscious experiences and are therefore believed to offer clues at how human consciousness works. One such illusion, the color phi phenomenon is marked by an anomalous time reversal of what is shown to test subjects and what they report to have seen. The color phi phenomenon has been a prime example to illustrate that conscious experience, instead of being merely a recording of the facts as they happen, involves both pre- and post-constructive processes resulting in a coherent narrative. We provide an alternative interpretation by showing that all aspects of this phenomenon can be adequately modeled in small neural networks. These networks are composed of simple leaky integrator neurons, emphasizing the importance of taking into account dynamical behavior. We illustrate how anomalous temporally-reversed perception of stimuli can appear based exclusively on bottom-up dynamics. Since the color phi phenomenon emerges as a feature of dynamical processing in our model, we conclude that it may explain less about human consciousness than previously thought and that it could be nothing more than a feature of dynamical processing in the brain.

Grant titled "Consciousness: The Radical Plasticity Thesis", SH4, ERC-2013-ADG. The funders had no role in study design, data collection and analysis, decision to publish, or preparation of the manuscript.

**Competing interests:** The authors have declared that no competing interests exist.

## Introduction

The color phi phenomenon (CPP), Fig 1, is perhaps one of the most baffling visual illusions known [1]. In the experiment, two different colored dots are shown in rapid succession and at some distance from one another. For certain intervals and spacings, which depend on the viewer, the perception is that of the first dot moving to the position of the second dot and changing to the color of the second dot somewhere along the perceived path. The perception of a mix of the colors of the dots is also possible. An interactive demonstration of the effect can be found online [2]. Clearly, the CPP is comprised of two separate effects. First, there is the phi phenomenon of perceived movement, which has been described since at least 1912 [3]. A swiftly moving "ghost" presence is perceived, located in between the positions of the discretely presented dots. The phi phenomenon has been the basis for moving pictures. The second part and the more striking anomaly of the CPP is that the color change is perceived before the second dot at the final position is. In the words of Herzog *et al.* [4]: *"The conscious percept cannot have formed in a time-ordered fashion, but must have been constructed retrospectively"*. The importance of the CPP is that it shows us that our conscious perception does not necessarily preserve the chronological order of the stimuli, nor does it need to correspond to actual applied stimuli.

The study of visual illusions such as Hermann grids and Mach bands which are direct illustrations of the existence of receptive fields, can offer important insights into our neural architecture [5]. Visual phenomena that are time-involved rather than static can hint at what kind of causal connections, feedforward, horizontal or feedback, exist between different brain areas [6]. In Ref. [7], it is argued that the receptive field of neurons in the visual cortex is mainly determined by feedforward processing, while feedback connections can modulate responses in V1 corresponding to perceptual grouping. In this way, the activity of feedback connections may correlate to perceptual awareness. However, as Bachmann [8] argues, it is difficult to distinguish experimentally whether top-down signaling to lower earlier activated feature-encoding neurons is due to higher level activation that is specific to target features or originates from non-specific activation linked to general arousal or expectation modulation that was initiated by feedforward non-feature-encoding reticulo-thalamic feedforward connections. In the majority of cases, top-down processing may be influenced by both specific and non-specific feedback.

Many illusions similar to the CPP exist that demonstrate that our perception is flawed when time is involved. For example, in the flash-lag illusion, the perception of the position of a

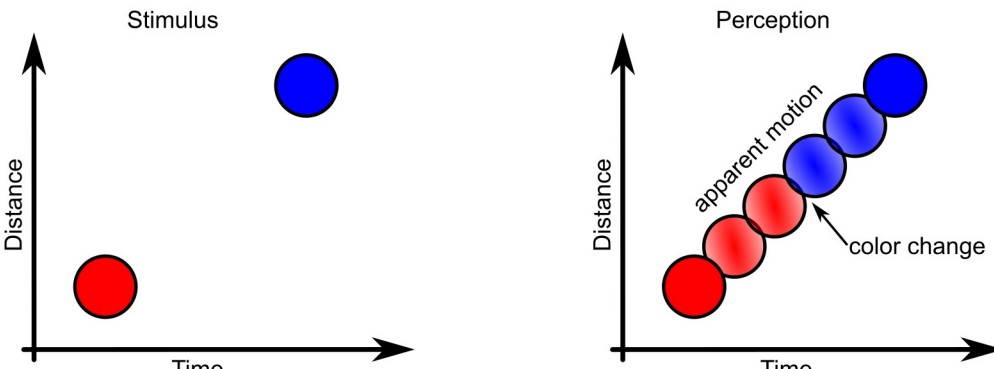

**Fig 1. The color phi phenomenon.** Two dots of different color are shown in rapid succession. For certain spatial and temporal distances, the perception is that of the first dot moving to the position of the second dot while abruptly changing color somewhere along the path.

moving object at an instant in time that another object is briefly shown, is systematically advanced relative to the true position [9]. The cutaneous rabbit illusion [10] is a tactile version of the phi phenomenon: taps or electrical pulses near the wrist and elbow in rapid succession create the feeling of a sequence moving up the arm, without there being any stimulus applied in between. The cutaneous rabbit illusion is accurately modeled by assuming perception follows a Bayesian inference process, based on a prior expectation of the stimulus having a low speed and uncertainty about its position [11, 12]. This results in an apparent distance contraction between the stimuli, perceptually leading to the rabbit hops. Given their success, these statistically based models add strength to the view of the brain as a (Bayesian) predictive inference machine [13], however they do not incorporate or highlight the underlying physiological connectionist basis.

In this article, we follow an approach that is more in line with O'Brien [14], in that perceptual awareness is seen as coinciding with at least temporally stable states or activation patterns existing in a connectionist system. Such an approach requires that phenomena such as the CPP are demonstrated and explained in a connectionist fashion, that is as existing in a neural network built up from elementary (artificial) neural units which have properties that are understood or accepted as being of similar nature as those of actual neurons in the brain. Most importantly, all these units operate continuously in parallel, hence the naming parallel distributed models (PDPs). Cohen *et al.* [15] presented a PDP account of the Stroop effect in a strictly feedforward dynamical model. Interestingly, their model had "attention" modulating inputs to steer the network's task to respond to either the word or the color it was presented. Such attention-modulating input would in reality clearly be a part of a top-down feedback path, even if it is not explicitly derived from output activations present in this network. Since this was a feedforward network, it does not have memory obtained from attractor states. In contrast, Dehaene *et al.* [16] present a dynamical model that relies heavily on both short and long distance feedback between individual simulated thalamo-cortical columns. Interaction between these columns, which consist of integrate-and-fire units, allows them to simulate the attentional blink phenomenon. Conscious reportability is seen as caused by the synchronization or locking of several simulated columns in a global activity pattern. The apparent similarity across modalities between the cutaneous rabbit illusion and the phi phenomenon suggests a common underlying mechanism which, in a connectionist view, could be supported by a common neuronal microcircuit. This motivates why the study of archetypal (dynamical) microcircuits that demonstrate comparable abstractions of these effects is important.

In this article, we offer fully detailed and testable connectionist models which demonstrate stimulus-response order-of-arrival reversals, as well as "ghost" perceptions. We do so by employing a rather crude neuron model, that nevertheless has all the necessary properties an artificial neuron requires to be able to perform general computation. By choosing to position ourselves at a well-chosen distance from more biologically informed models, we are able to illustrate general principles.

With subjective experience changing from moment to moment, it is clear that consciousness itself takes time. Therefore, dynamical systems are an appropriate basis for modeling that offers insight in *how things might be the way they are* as, argued by Cleeremans and French [17], that is all modeling does. Differential equations (DEs) and their discrete-time counterparts difference equations form the natural language of dynamical systems [18]. Coupling of the same type of DE that describes an aspect of neural activity is how dynamical connectionist networks are built.

The CPP received some fame for being the centerpiece in an argument formulated by Dennet against the existence of a "Cartesian theater", an informational finish line in the brain where the order-of-arrival coincides exactly with the order of conscious perception [19].

However, we argue, no matter in which manner the anomalous perception such as that of the CPP comes about, at some point, somewhere in the brain, there must exist an activation that reflects this subjective perception, if one is to accept that perception itself is physical. Therefore there must exist some neuronal circuit that transforms the applied stimuli in objective activations that in turn either *are* or cause the anomalous perception. Our goal here is to demonstrate that these phenomena can be modeled in the most simple way as generic dynamical effects. Note that we choose in this article to focus on mechanisms. Therefore we specifically do not correlate our models with results of behavioral experiments yet. Also, we remain agnostic as to the existence of feature vectors that include time labels as proposed in Herzog *et al.* [4]; our models do not require them as far as the CPP goes, but could be made compatible with that idea.

The remainder of this article is organized as follows. In the next section we introduce the dynamical neuron model we employ. Thereafter we describe a four-neuron network that illustrates the stimulus-response arrival order reversal. We introduce a model for the phi phenomenon, the operation of which depends on long distance feedforward "primer" connections. We then model the complete CPP using an echo state network (ESN), a recurrent dynamical neural network built from a large number of randomly connected toy model neurons. Finally, we summarize our results and propose ways in which we see this work continue.

## Toy model neuron

Dynamical neuron models fall into two broad categories: spiking [20, 21] and non-spiking. Although spiking neuron models, which are inherently time-dependent, come closer to biological reality, they also form an extra hurdle since neural coding, apart from rate coding, is not completely understood and still actively researched. We choose here to go with a dynamical version of the simplest of all non-spiking neuron models, the perceptron [22]. Specifically, we aim to develop our arguments starting from three well accepted basic prepositions:

- The computational power of a neuronal systems (brains) stems from the connection of many, nearly identical and relatively simple units, called neurons.

- Neuronal signals are limited in amplitude and rate. Neurons and neuronal systems can saturate and thus are inherently nonlinear. If stimuli $a$ and $b$ elicit responses $f(a)$ and $f(b)$ respectively, then the simultaneous application of $a$ and $b$ does in general not result in response $f(a + b)$.

- Neurons and neuronal systems have a certain slowness to them. They cannot respond immediately to a change in input stimulus.

The first two properties give traditional artificial feedforward neural networks their general computational abilities, under certain mathematical conditions. The third property brings time and dynamical systems into the discussion. We start from the well-known perceptron neuron:

$$y = f\left(\sum_{i=1}^{N} w_i x_i + b\right),\qquad(1)$$

in which the synaptic weights $w_i$ determine the input strength of input $x_i$ and $f$ is a nonlinear activation function. The weights $w_i$ can be positive (excitatory) or negative (inhibitory). In general, a constant bias $b$ is also included. A straightforward dynamical extension is given by the

following ordinary differential equation:

$$\tau\frac{dy}{dt} = -y + f\left(\sum_{i=1}^{N} w_i x_i + b\right),$$

(2)

which is also known as the leaky-integrator neuron [23], not to be confused with the spiking leaky integrate-and-fire neuron. Here, we will just call it a toy model neuron, since that is what it is: an abstract entity with minimal neuron-like properties. The derivative $dy/dt$ is the velocity at which the output $y$ changes. Here $\tau$ is a timescale, with higher $\tau$'s meaning slower neurons. Note that for $\tau \to 0$ the response $y(t)$ of Eq (2) equals that of Eq (1). The same is true for $t \to \infty$, with a constant input $x$, which can be seen from setting $dy/dt$ to zero. Thus Eq (2) represents a "slow" perceptron. The nonlinear activation function is usually a hyperbolic tangent, with output values ranging from −1 to 1 or a sigmoid $f(x) = 1/(1 + \exp^{-x})$, having a range of 0. . .1. Both are monotonic functions: increasing the stimulus strength leads to an increased response, up to saturation. Eq (2) can be interpreted as the average activity or firing rate of a single neuron as well as the activity of a larger neuronal (sub)structure. Our argument is that any phenomenon that can exist in a small network of these crude dynamical neurons, can also be present in the vastly more complex brain. Models built using this neuron then show which computational pathways might exist. Note that in Eq (2) all variables, including time $t$, are considered de-dimensionalized, as this simplifies notation [18].

## Example: Backward temporal masking

Merely as a starting point and to illustrate what is possible with this model, consider the single-neuron circuit shown in Fig 2A. We will show it can serve as an elegant model for backward temporal sensory masking. The reader who is familiar with nonlinear dynamics can skip the remainder of this section. The circuit in Fig 2 is represented by:

$$\tau\frac{dy}{dt} = -y + f(w_1 y + w_2 x)$$

(3)

in which the parameters are chosen as: $\tau = 1$ and $w_1 = w_2 = 1.5$. Here, for simplicity, the activation function is chosen to be a piece-wise linear function, of which the shape is reminiscent of a hyperbolic tangent:

$$f(x) = \begin{cases} -1 & \text{if } x < -1, \\ x & \text{if } -1 \leq x \leq 1, \\ 1 & \text{if } x > 1. \end{cases}$$

(4)

Plotting the right hand side of Eq (3), Fig 2B (top trace), for input $x = 0$, shows three points where $dy/dt$ is zero. In other words, this system has three equilibrium points. A small positive push from the origin $y = 0$, will result in $dy/dt > 0$, thus the system evolves to more positive $y$ values, until it reaches the right-hand side equilibrium $y^* = 1$. Here it remains, because a small positive nudge towards values $y > 1$ results in a negative change $dy/dt < 0$. The treatment for the left hand equilibrium at $y^* = -1$ is similar, showing it is also stable. Thus the system has a single unstable equilibrium in the center $y^* = 0$ (indicated by an empty dot) and two stable equilibria at $y^* = \pm 1$ (filled dots). The latter are the results of the above-unity positive feedback weight $w_1$.

As discussed in the Introduction, stable attractor states in the brain, at least temporary ones, have been hypothesized as the basis for conscious percepts [14, 24]. We emphasize

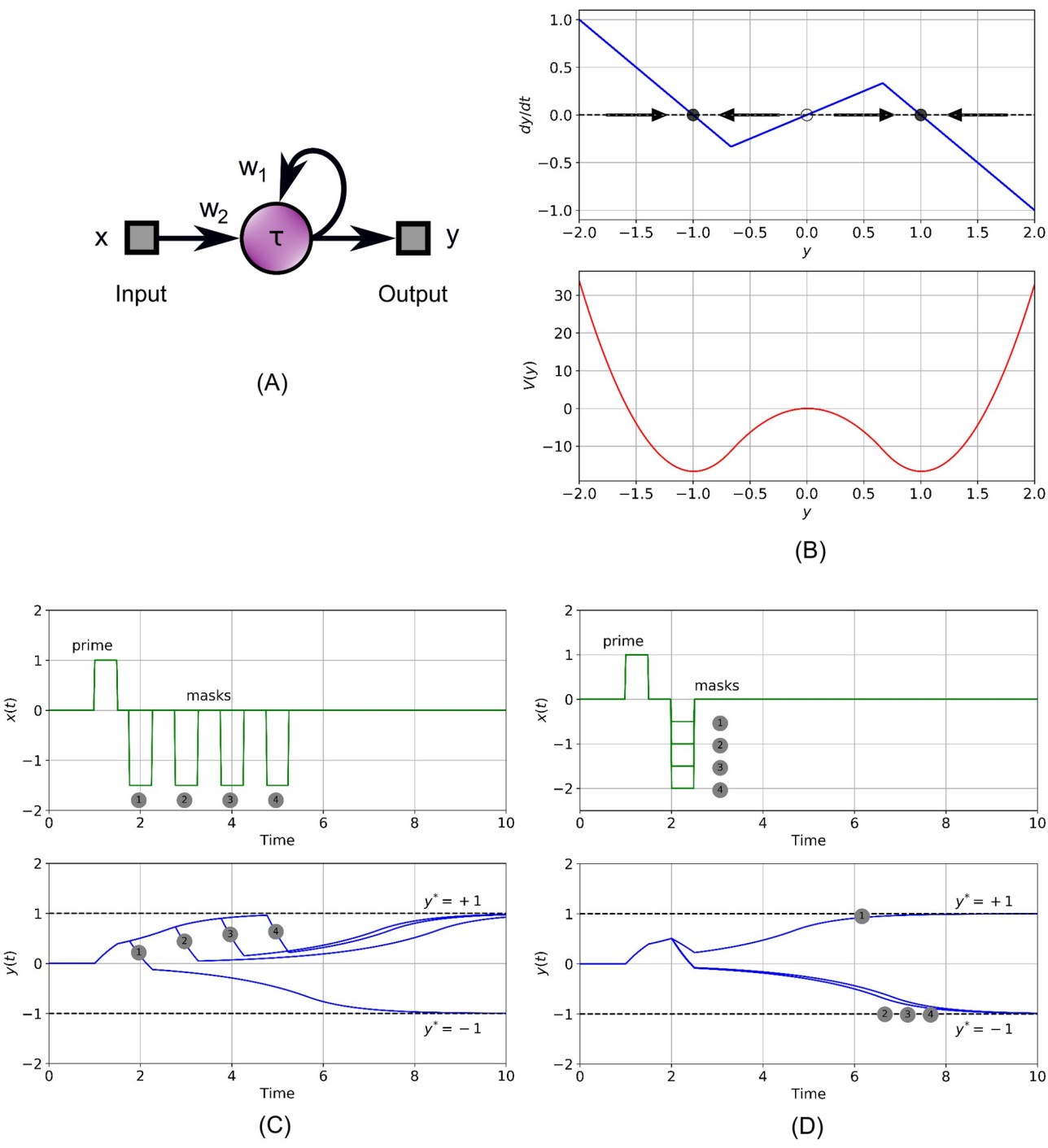

**Fig 2. Backward temporal masking with a single dynamical neuron.** (A) Neuron circuit with excitatory self-feedback, Eq (3), illustrates backward temporal sensory masking. $w_1 = w_2 = 1.5$, $\tau = 1$. (B) Positive feedback creates two stable equilibria at $y^* = \pm 1$ (top trace). These form the minima of a gradient-descent function $V(y)$ (bottom trace). (C) Depending on the interval between a priming and a countering stimulus of opposite polarity, masking can occur. (D) The effectiveness of the masking stimulus also depends on its amplitude.

that although the example given in this section is original work that to the best of our knowledge has not been shown earlier, it is hardly original in spirit and only serves as an introductory example. Note that technically, besides stable states, attractors can also include stable oscillations known as limit cycles and chaotic oscillations, which take in extended regions

rather than isolated fixed points, in the phase space of a dynamical system. Mathis and Mozer [24] were able to correlate several priming experiments to a discrete-time model. Their model is based on a radial basis function (RBF) attractor network, instead of a specific type of (networked) artificial neuron, and falls into the category of gradient descent systems. Formally, in a (multivariable) gradient descent system, the change of the system state $\vec{y}(t) = (y_1(t), \ldots, y_K(t))$ can be written as:

$$\frac{d\vec{y}}{dt} = -\vec{\nabla} V(\vec{y}) + \vec{x}(t), \tag{5}$$

where $\vec{\nabla}$ is the Nabla operator: $\vec{\nabla} V(\vec{y}) = \left( \frac{\partial V}{\partial y_1}, \ldots, \frac{\partial V}{\partial y_K} \right)$. When undriven by a stimulus $\vec{x}(t)$, the state evolves downward towards a minimum of a scalar function $V(\vec{y}) : \mathbb{R}^K \to \mathbb{R}$, much like a golf ball rolling downward to the lowest point of a valley it happens to find itself in. The influence of the input stimulus is such that it can drive the ball, representing the state of the system at every point in time $t$, closer or farther away from specific local minima of the $V(\vec{y})$, corresponding to influencing both which perception the stimulus elicits and the time it takes to do so. Eq (3) is actually also a gradient system, albeit a one-dimensional one. This is shown in Fig 2B (lower trace). The piece-wise linear activation function Eq (4) allows integration of the top trace, so the undriven system can be written as $dy/dt = -dV(y)/dy$. This leads to the piece-wise parabolic gradient function $V(y)$, shown in Fig 2B (bottom trace). The local minima of $V(y)$ are at the stable equilibria $y^* = \pm 1$ of the system. The maximum at $y^* = 0$ is an unstable equilibrium. An animation in the supporting information, S1 Animation, shows how the stable "pockets" or valleys are created as the positive feedback $w_1$ increases.

In above setting, conscious perception coincides with a state sufficiently close to an equilibrium of a, not necessarily explicitly described, gradient system. Thus this exemplary system is capable of two percepts $y = +1$ and $y = -1$, when started from $y = 0$ at $t = 0$. The outcome depends on the stimulus $x(t)$ during the time in which the dynamical state $y(t)$ changes. Note that short-lived or transient states can also represent distinguishable conscious percepts, if they can cause distinguishable stable states in subsequent processing units exhibiting longer, possibly infinite, memory.

In Fig 2C (top trace), we show how this simple one-neuron system is useful in a computational version of a stimulus onset asynchrony experiment. Here, a priming stimulus of unit strength and duration 0.5 starts the system on a path to experience the positive percept (bottom trace). Some time later, during the transient, a single negative masking stimulus of amplitude −1.5 is administered. Depending on the time elapsed between the priming and masking stimulus, the system ends up in either one or the other equilibrium. The self-reinforcing feedback can be interpreted as the system making a top-down prediction: in the absence of new information, it will move towards the state that coincides with the direction that the initial input pushed it towards. Contrasting information, in the form of input steering the state in the opposite direction can be taken into account, however it must overcome the already accumulated "proof" which, especially in later stages, becomes more difficult to do. That is why in this example, even though the masking stimuli are stronger than the prime, only the one closest in succession to the primary stimulus is able to mask it successfully. In Fig 2D we repeat this simulation for different amplitudes of the masking stimulus but for a fixed time post-stimulus. As expected, the strength of the mask influences its effectiveness. Note that even if a masking stimulus is not effective in blocking the prime, it can still delay the response. Thus this simple model predicts that effective backward sensory masking depends on both amplitude and elapsed time since the primary stimulus. We have included an animation in the supporting information, S2 Animation, showing how the state moves towards a minimum of the gradient

function. This model, given merely to exemplify the merits of dynamical systems, is of course too simple to catch all scenarios. For example blanking, leading to no perception, is nearly impossible here since the mask must cancel the primary stimulus exactly. This is due to the unstable equilibrium in the origin. Adding noise to Eq (3), such as is present in a drift model, randomizes the outcome and leads to a probability distribution that can be compared against results from actual experiments. Note that a non-dynamical sum-saturate neuron can be used to show (discrete-time) dynamical effects such as memory when it is used in the context of a recurrent network. An example is given by the simple recurrent network [25], where due to discrete state updates previous inputs play a role in determining the current output.

In the next section, we show a small network of toy model neurons that illustrates the first part of the CPP, namely how perceptual states and their respective causal stimuli can appear to be chronologically reversed under certain circumstances.

## A dynamical model for stimulus-response order of arrival reversals

In Fig 3A, we show a neuronal circuit with four neurons of the type described by Eq (2). Two inputs $x_a$ and $x_b$, feed into two chains each having two neurons, top: $a_1$, $a_2$, bottom: $b_1$, $b_2$. The respective outputs of the chains, $y_{a2}$, and $y_{b2}$, are taken to coincide with the perception of the corresponding stimulus, when above a chosen threshold value. Inhibitory (negative) "long distance" connections cross couple the inputs to the opposite chain. Formally, the circuit is represented by:

$$
\begin{aligned}
\tau_{a1} \frac{dy_{a1}}{dt} &= -y_{a1} + f(w_{xa,a1}x_a + w_{a1,a1}y_{a1} + b_{a1}), \\
\tau_{a2} \frac{dy_{a2}}{dt} &= -y_{a2} + f(w_{a1,a2}x_a + w_{a2,a2}y_{a2} + w_{xb,a2} + b_{a2}), \\
\tau_{b1} \frac{dy_{b1}}{dt} &= -y_{b1} + f(w_{xb,b1}x_a + w_{b1,b1}y_{b1} + b_{b1}), \\
\tau_{b2} \frac{dy_{b2}}{dt} &= -y_{b2} + f(w_{b1,b2}x_a + w_{b2,b2}y_{b2} + w_{xa,b2} + b_{b2}),
\end{aligned}
\tag{6}
$$

where we chose the parameters as: $\tau_{a1} = \tau_{b1} = 2$, $\tau_{a2} = \tau_{b2} = 10$, $w_{xa,a1} = w_{xb,b1} = 1$, $w_{a1,a2} = w_{b1,b2} = 1$, $w_{xa,b2} = w_{xb,a2} = -2$, $b_{a1} = b_{a2} = b_{b1} = b_{b2} = -0.5$. As activation function we choose:

$$
f(x) = \begin{cases} 0 & \text{if } x < 0, \\ x & \text{if } 0 \leq x \leq 1, \\ 1 & \text{if } x > 1. \end{cases}
\tag{7}
$$

The threshold of perception was chosen as 0.5, equal to midway of the range of the nonlinear function $f$. Conceptually, the existence of this arbitrarily chosen threshold does not have to be problematic, as subsequent structures that take the outputs of the circuit may be tuned to have an all-or-nothing response based on this threshold. In Fig 3B, we show how differently spaced input stimuli (of width $\delta = 10$), cause different response times $RT_a$ (red) and $RT_b$ (blue). For both large positive and negative input intervals, $\Delta t_{in} = t_{in,b} - t_{in,a}$, the response arrival order is unaffected, i.e. first in first out. However, for $\Delta t_{in} \approx -12$ to 12, we see $\Delta t_{out} = t_{out,b} - t_{out,a}$ (black) has the opposite sign of $\Delta t_{in}$, i.e. last in first out. Without the reversal effect, the green dashed line would be followed. Note that if both inputs are activated simultaneously, then both RTs are increased, however the interval between the responses remains zero.

Fig 3C shows how the model behaves if we vary both the amplitudes of the input stimuli and the interval between them. Here, the sum of the amplitudes is kept at 1.5 and their ratios

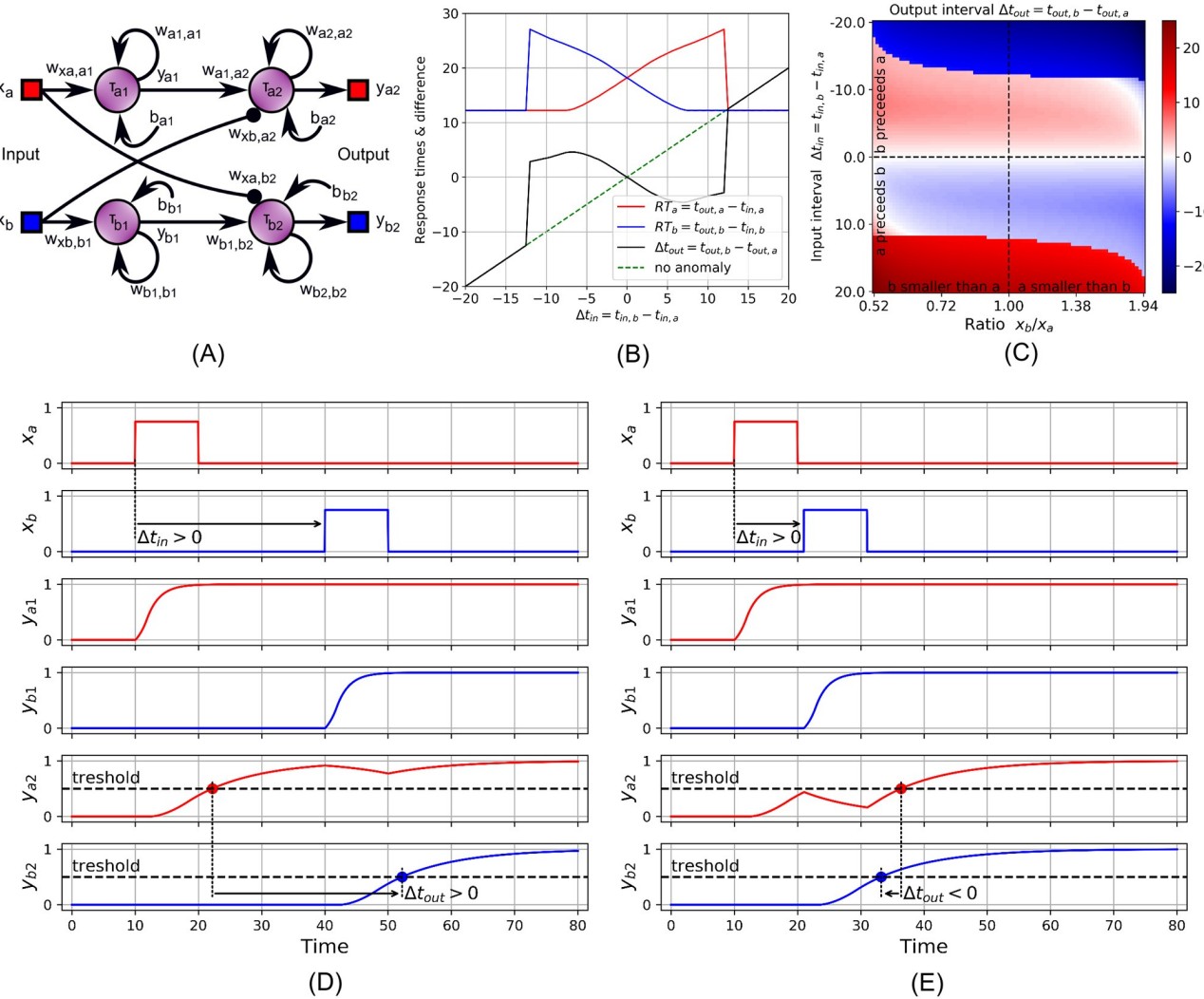

**Fig 3. A four-neuron model for demonstrating stimulus-response arrival order reversals.** (A) The model. See below Eq (6) for parameter values. (B) A Scan over stimuli intervals $\Delta t_{in} = t_{in,b} - t_{in,a}$ showing the influence on response times $RT_a = t_{out,a} - t_{in,a}$ (red), $RT_b = t_{out,b} - t_{in,b}$ (blue) and the response time interval $\Delta t_{out} = t_{out,b} - t_{out,a}$ (black). For a certain central region of the input interval, there exists a reversal of the order in which responses happen, compared to the order in which the stimuli were applied. (C) Scan over stimuli intervals and amplitude ratios, illustrating how temporal and amplitude-related stimuli characteristics influence their temporal perception. (D) Detailed signals for $t_{in,b} \gg t_{in,a}$ such that the output order corresponds to the input order. (E) Detailed signals in the input-output reversal regime, where $\Delta t_{in} > 0$ and $\Delta t_{out} < 0$.

are varied. In Fig 3C, the lighter colored central part marks output intervals that are temporally reversed relative to the input interval. There is a tendency, within certain limits, that the more stronger a later arriving stimulus is, the easier it can overtake an earlier one. Thus the model demonstrates how both temporal and amplitude-related characteristics of complex stimuli influence their temporal perception in a non trivial way.

Fig 3D and 3E illustrate this process in detail. The input stimuli trigger a latching value at the output of the first neurons, $y_{a1}$ and $y_{b1}$, in both chains. These in turn drive the subsequent stages. This however takes some time due to the slower timescale of neurons $a_2$ and $b_2$. If the input $x_b$ is activated while the output value $y_{a2}$ is still accumulating, Fig 3E, then the inhibitory synapse slows down the $a$-branch response. This leads to $y_{b2}$ crossing the threshold before

$y_{a2}$, even if stimulus $x_a$ was present before $x_b$ was. Since the circuit is symmetrical along the branches, swapping the input stimuli leads to swapped output signals.

The circuit of Fig 3A clearly illustrates how dynamics analogous to anomalous order-reversed perception can be found in small dynamical networks. The key ingredient is the set of cross coupled inhibitory connections which jump forward in the chains, such that information that arrives at a later time can intercept partially processed information at the opposing chain as it flows from input to output. Loosely speaking, we make an exchange of time for space.

In the next section, we illustrate how the second aspect of the CPP, an apparent movement or "ghost" perception, can be found in a similar small network motif.

## A dynamical model for the phi phenomenon

As mentioned, the color phi phenomenon has two aspects. In this section we explain how the appearance of a ghost percept, spatially and temporally placed in between percepts that correspond to actual stimuli, might be understood as a dynamical priming effect. In Fig 4A, we show a toy model neuron circuit consisting of 15 units, divided in three chains $a$, $b$ and $c$. It represents a very crude one-dimensional visual system having three pixels. Stimuli on the respective inputs $x_a$, $x_b$ and $x_c$, Fig 4B (top), lead after some propagation time to perceptions as indicated by the outputs $y_a$, $y_b$ and $y_c$ (bottom) being above a fixed threshold value. Each chain couples to the adjacent chain(s) via weak long distance feedforward synapses, shown as dotted connections in Fig 4A. These are driven by primer neurons $p_a$, $p_b$ and $p_c$, Fig 4B (middle), that have a fast activation and a slow decay, the latter due to the strong self-feedback and lack of negative bias. In Fig 4B (middle), the dotted line shows the combined input of the primers into the final neuron of the $b$-chain. As long as the interval between the stimuli on the $a$ and $c$ chain is sufficiently large, their combined priming signal does not exceed the activation threshold of the $b$-chain, Fig 4C. However, when the stimulus on $c$ swiftly follows the one on $a$, as in Fig 4D, priming exceeds the threshold and elicits a response at $y_b$. As in the preceding section, we find a timing anomaly: $b$ is initially "perceived" after $a$ and before $c$, here close to $t = 115$. An animation showing the situation for intermediate $a$-$c$ stimulus intervals can be found in the supporting information, S3 Animation. Since the network is symmetric, the same effect exists when the stimuli are applied in reverse.

One may wonder how this simple model can be interpreted within a biological context. If a moving object, e.g. a prey animal, is seen at some position, say $b$, it is natural to assume that it will be moving to an adjacent position, here either $a$ or $c$, next. The priming signals thus perform the function of expectation. Priming for the adjacent positions helps reduce response times, and increases the chance of catching that prey. Likewise, if an object is perceived as quickly changing its location from $a$ to $c$, Fig 4D, then by assuming movement and thereby inferring that a path exists between the positions, the organism is potentially given useful information about the environment. If the objects are perceived with a greater temporal interval at positions that are not adjacent, as in Fig 4C, they are likely not the same and the existence of a pathway cannot be inferred. Thus this simple model demonstrates that perception, even when caused mainly by feedforward neuronal pathways, can in part be prediction [13]. Note that these interpretations are meant to place the abstract model of Fig 4 in a more understandable context, and by no means do we want to state physiological facts.

Clearly, this simple hand assembled model fails in two ways. First, in Fig 4D, there is overlap between the $b$ and $c$ percepts. Second, if $x_a$ and $x_c$ are stimulated at the same time, $y_b$ will also be seen, while in an actual experiment only the two outer position dots would be perceived simultaneously. These shortcomings could perhaps be alleviated at the expense of increased model complexity, by using inhibitory connections from the inputs to the outputs. Similar to

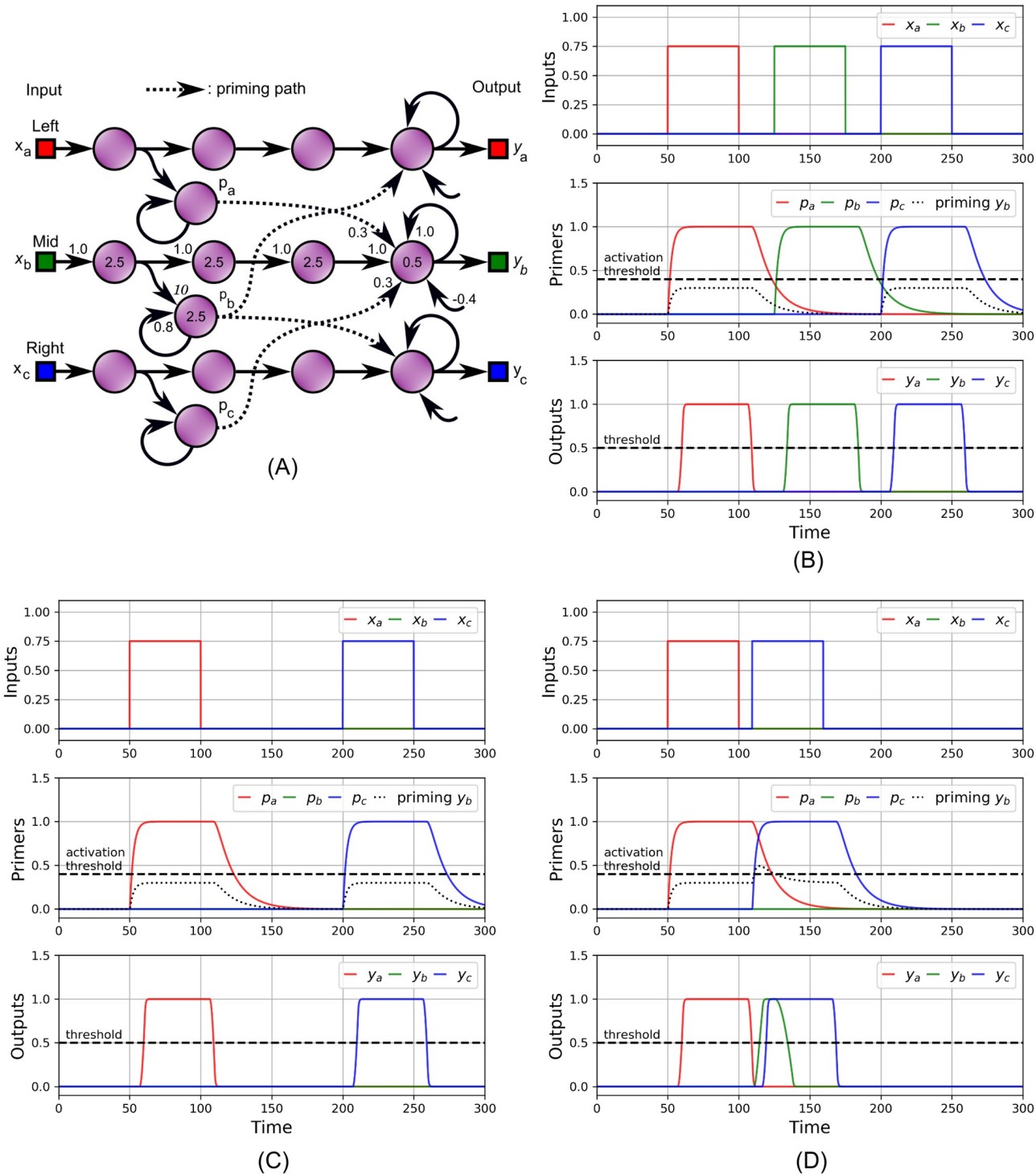

**Fig 4. An abstract one-dimensional three-pixel visual system, illustrating the phi phenomenon.** (A) The circuit. Outputs $y_a$, $y_b$ and $y_c$ separately respond to inputs $x_a$, $x_b$ and $x_c$ respectively, corresponding to perception when above a certain threshold. Weights and $\tau$-values of the $a$ and $c$-chains are identical as those indicated in the $b$-chain. (B) Long distance weakly coupled feedforward priming units $p_a$, $p_b$ and $p_c$ (middle graph) with fast activation and slow decay enable sensitivity to recent inputs to be carried to adjacent output units. (C) If the input pulses on $x_a$ and $x_c$ are temporally sufficiently separated, the decayed priming signals $p_a$ and $p_c$ are unable to evoke a response from the middle output $y_b$. (D) When inputs $x_a$ and $x_c$ are stimulated adequately fast after one another, the priming signals combine and a ghost percept on $y_b$ appears. This ghost happens after $y_a$ and before $y_c$ cross the threshold in opposite directions. The ghost appears without the presence of an input stimulus on $x_b$.

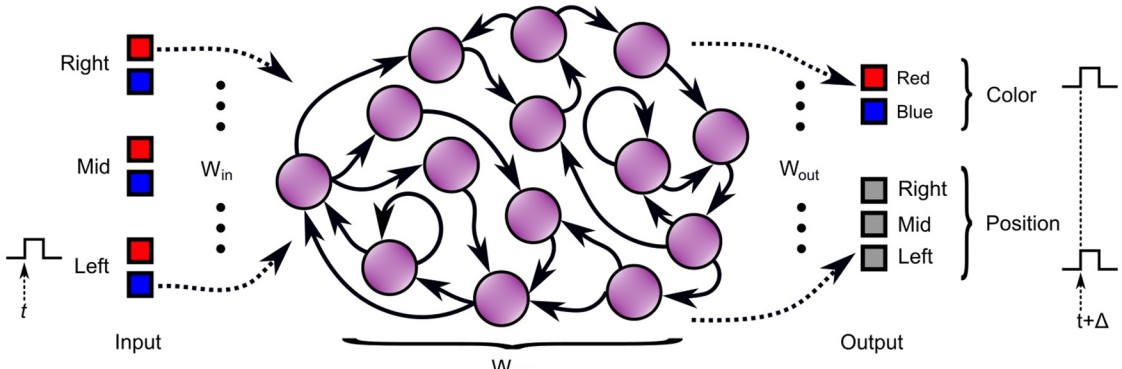

**Fig 5. Illustration of the color phi phenomenon via an echo state network.** A large randomly connected network of neurons, also called a reservoir, interacts in a complicated and unknown way in response to an input stimulus. The connections between the neurons in the network remain fixed for the duration of the experiment. The desired output response is constructed from a linear combination of the responses of the individual neurons that form the network. After training, the ESN can generalize to novel input data.

the case of the order reversal model, as shown in Fig 3C, the stimulus characteristics influence whether or not a phi phenomenon happens. Our code S1 Data contains a script that plots the duration of the ghosts as a function of stimulus interval and amplitude ratio.

In the next section, we show a different approach to finding such a model and demonstrate that the CPP could be understood as nothing more than a side effect of dynamical processing.

## The color phi phenomenon in an echo state network

The models presented in the previous sections illustrate both aspects of the CPP as discussed. Unfortunately, there seems to be no straightforward way to combine their structure into a single CPP model. Therefore, we turned to neural network paradigms to find such a model.

Specifically, we employ an echo state network (ESN) [26]. For completeness, we briefly explain the philosophy behind the ESN, after which we formally introduce our model. ESNs where envisioned to circumvent the problem of slow backpropagation learning found in traditional multilayer neural networks [27, 28] and to be able to exploit the computational abilities of high-dimensional nonlinear physical substrates in which the constituents have a fixed and often unknown interconnectivity. More in-depth discussions of ESNs, part of the larger group of reservoir computing methods, can be found in Refs [29–31].

Instead of adjusting individual weights between neurons, such that the network itself performs a certain prescribed function, in an ESN, Fig 5, one starts from a large recurrent network, also called a reservoir, in which the weights between the network's neurons or nodes are chosen at random, within certain constraints, and kept fixed throughout the experiment. The presence of recurrent connections implies the existence of time scales or memory on top of those offered by the neurons' individual dynamical behavior. An input signal is connected, usually also via randomly chosen weights, and excites the network. For our purposes, the ESN will have $3 \times 2$ inputs corresponding to left, middle and right positions in two colors, blue and red, similar to the abstract one-dimensional visual system described in the previous section. The complicated time-dependent response or "echo" in the network caused by the time-varying input is then recorded. Mathematically, the input signal is non-linearly projected into the high-dimensional phase space of the dynamical network. The training phase of the ESN consists of constructing the desired output response from a linear combination of the response

signals of the individual neurons. The weight values or coefficients of this linear combination are collectively known as the readout layer. Mathematically, the readout layer forms a down-projection from the phase space of the network onto a lower dimensional output space. The main idea behind ESNs is that as long as the network shows what is loosely referred to as "rich dynamical behavior" in response to the input, chances are good that a projection exists that maps the dynamical response of the network sufficiently close to the desired output. The calculation of the readout layer is a simple linear regression, which is computationally relatively cheap and fast. In this case, we have five outputs: three for the position and two for color. As such, the quite easy task of the ESN is to classify a single active input. Furthermore, when several inputs are active at the same time, we demand that all outputs must remain inactive. Since the ESN is a dynamical system, it will show transition effects between different input stimuli. After training, the ESN will be tested for the CPP as further outlined below. It is crucial to remark here that the network is not trained to show the CPP explicitly. As explained in detail in the next section, the ESN consists of a discrete-time version of the toy model neurons, Eq (2).

Interpreted as generating perceptions, the readout layer is the place where "it all comes together", i. e. the Cartesian theater. Note that it is not correct to interpret the readout layer as simply being a reporting tool for the network, a voice box so to speak, since without it, the computations that happen in the network do not get assembled into an active representation and remain inert. Alternatively, the outputs of the readout layer could be seen as a behavioral response. The reservoir itself could be extended to have specialized readout neurons, the outputs of which perform exactly as the readout layer.

## ESN outline

In this section we formally describe the ESN we use to demonstrate the CPP. Starting from a discretization with timestep $T_s$, setting $t = nT_s$, we replace differentials with differences as:

$$\frac{dy}{dt} \equiv \frac{y(nT_s) - y(nT_s - T_s)}{T_s},\tag{8}$$

such that the toy model neuron Eq (2) takes on the form of a recurrence equation:

$$y_{n+1} = (1 - \alpha)y_n + \alpha f\left(\sum_i^N w_i x_i + b\right),\tag{9}$$

in which $n$ is the discrete timestep and the neuron timescale is given by $\alpha = (1 + \tau/T_s)^{-1}$. Compared to the artificial neurons used in Cohen [15], in Eq (9) the differentiating action and the saturating function are swapped, however their steady-state output remains the same. The ESN state update equation is functionally equivalent to what is found in Ref [23]:

$$\vec{x}_{n+1} = (I - A)\vec{x}_n + Af\left(W_{res}\vec{x}_n + W_{in}\vec{x}_{in,n} + \vec{b}\right),\tag{10}$$

in which $\vec{x}_n = (x_{1,n}, \dots, x_{N,n})$ is the network state vector at timestep $n$. In Eq (10), vectors are to be interpreted as column matrices. Furthermore, $W_{res}$ is the $N \times N$ network connectivity matrix, $W_{in}$ is the $N \times 6$ input weights matrix and $\vec{b} = (b_1, \dots, b_N)$ is a constant random bias vector. Vector $\vec{x}_{in}$ represents the six inputs as shown in Fig 5. In our experiments, we set the network size to $N = 200$ neurons. Network hyperparameters were chosen as suggested in Ref. [32]. The network connectivity matrix is constructed by drawing random numbers from a normal distribution $\mathcal{N}(\mu = 0, \sigma = 1)$ in which 80% of the entries is set to zero (sparsity 0.8). Then the matrix is scaled to obtain a spectral radius of 0.9. Similarly, the input matrix $W_{in}$ is constructed

from a normal distribution $\mathcal{N}(\mu = 0, \sigma = 0.5)$ with sparsity set to 0.8. The timescales of the individual neurons are drawn from a uniform distribution $\alpha_i \sim \mathcal{U}(low = 0.1, high = 0.3)$, and collected in a diagonal matrix $A_{i,i} = \alpha_i$. Here, $f$ is the sigmoidal activation function. Both during training and testing of the ESN, the network states in response to the input signal are collected in a state matrix $X$ of dimensions $K \times (N + 1)$. Each row of $X$ consists of the network state $\vec{x}_n$ at time $n = 1, \ldots, K$. The last column of $X$ is constant. The readout layer consists of a $(N + 1) \times 5$ matrix $W_{out}$. The output of the ESN is calculated as:

$$Y = XW_{out}, \tag{11}$$

where $Y$ is a $K \times 5$ matrix, of which each column corresponds to an output signal as shown in Fig 5. The $n$-th row of $Y$ corresponds to the output vector at timestep $n$, $\vec{y}_n = (y_{1,n}, \ldots y_{5,n})$. The readout layer $W_{out}$ is determined using a linear regression, such that the output $Y_{train}$ in response to a training input that causes network states $X_{train}$ comes close, in a least-squares sense, to a predefined desired output $Y_{desired}$:

$$W_{out} = (X_{train}^T X_{train})^{-1} X_{train}^T Y_{desired} \tag{12}$$

A well-trained ESN can generalize to previously unseen input data. We refer the reader to Refs [29–31], as further details about the ESN are not important for the remainder of this article.

## ESN experimental results

In this section we show how the CPP surfaces in the ESN we described above. As shown in Fig 6, the training input (top graph) consists of valid one-position/one-color stimuli, for which the desired output (third graph) is the corresponding position and color. From $n = 13000$ onward, the training stimulus consists of invalid input conditions in which multiple color and/or position inputs are active simultaneously. For these confusing stimuli, we require the network to not respond (keep all outputs at zero). A shift of 20 timesteps between input and desired output was allowed to give the network time to respond. After training, i.e. calculating the readout layer $W_{out}$, we can show the resulting actual output in response to the training input (bottom graph of Fig 6). The output is assembled from the complex network dynamics (first ten neurons shown in the second graph of Fig 6). Ideally, it should be identical to the desired output. An initial transient and some noise-like deviation is seen, however qualitatively the output comes close to what is desired.

It is vital for this discussion to emphasize that we do not train with input conditions that could lead to the CPP: transitions between the conditions left-to-right or right-to-left of opposite color without passing via an activated middle input are explicitly excluded from the valid training examples. Furthermore, input conditions are sufficiently spaced as to allow the network's internal dynamics to settle to a stable output. We are now ready to demonstrate that this neural network can undergo a color phi illusion. In Fig 7A (top), we apply several CPP input conditions, i.e. a stimulus that jumps from left-red to right-blue, with decreasing interval. Note that the left-red and right-blue input pulses never overlap. Ideally, the middle output would remain low (middle graph), corresponding to having not perceived anything anomalous. However, as the left-red to right-blue jumps get closer to one another, at $n \approx 3490$, as shown in the zoomed graph of Fig 7B (bottom graph), the middle output is briefly active above the threshold of 0.5 (dashed horizontal line), while the right output is still below this threshold. This corresponds to a ghost or phi phenomenon as elaborated on in the previous sections. More so, at the same time the blue output is also active, thus the output of the ESN

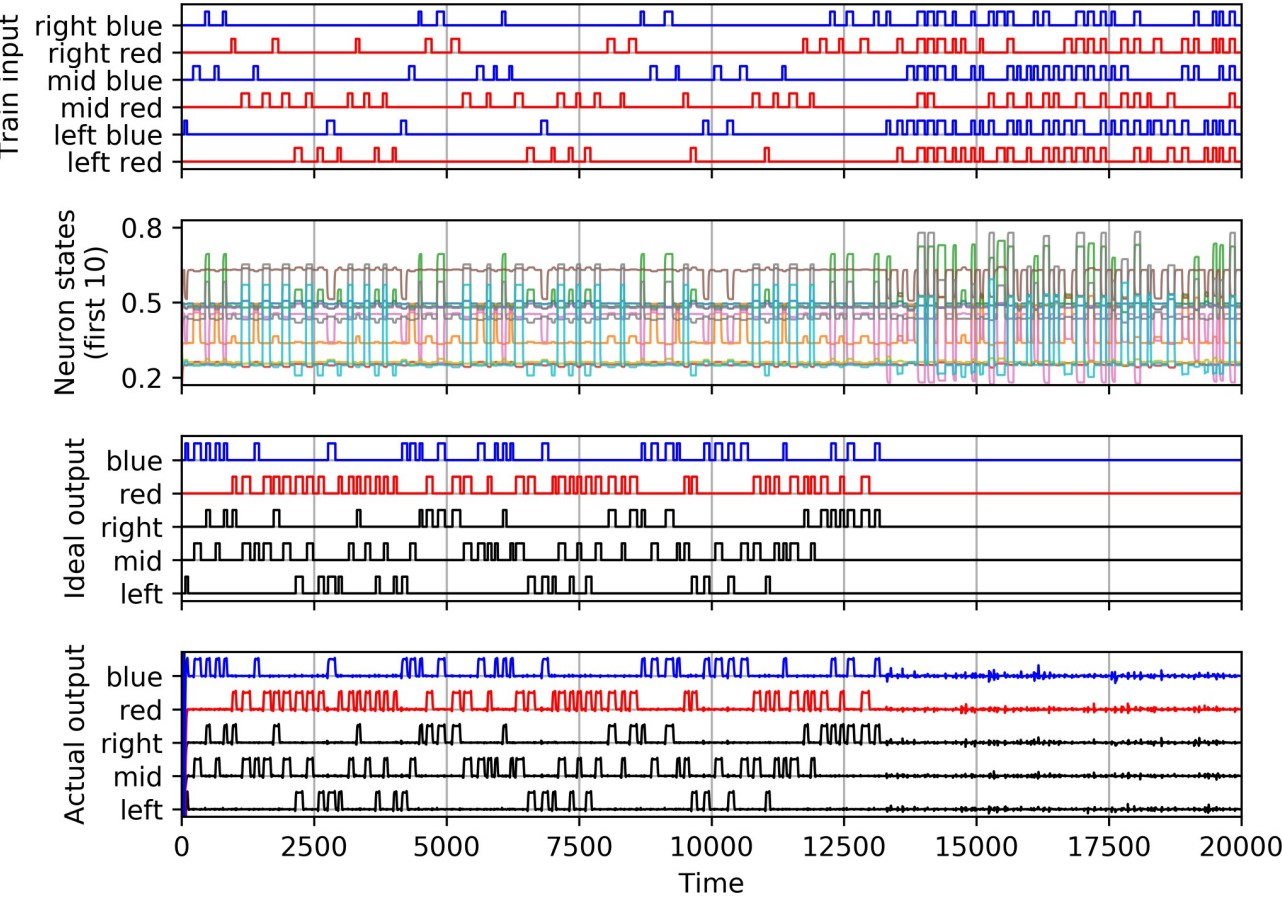

**Fig 6. ESN during training.** (top) Training input, (second graph) states of the first ten neurons, (third graph) desired output used to calculate the readout layer, (bottom graph) actual output calculated by the readout layer from the neuron states after training.

corresponds with the perception of this ghost in the middle position and in the final color *before* correctly perceiving the second applied stimulus of that color.

The dashed vertical line in Fig 7B (bottom graph) indicates the point in time where this condition first occurs. Thus the system portrays all the characteristics of undergoing a color phi illusion. The supporting information contains a short clip S4 Animation produced directly from the timeseries of Fig 7A, showing how the ESN "perceives" the CPP. The perception of the final transition from left red to right blue briefly shows a blue dot in the middle.

**Analysis.** Since the model is essentially a system of $N = 200$ coupled nonlinear difference equations, we cannot explain in detail how this behavior arises. It is obvious, however, that simple dynamical interactions give rise to the existence of the CPP here. Nevertheless, it is important to ask if there are any discernible differences between ESNs that do and those that do not show the CPP effect. In Fig 8 we present four network characteristics that we specifically checked to identify such differences. The plots are based on $N = 100000$ randomly assembled ESNs in total. In general, only a low fraction of the ESNs show the CPP. In our experiments this was 1.87%. Fig 8A shows the distribution of $\alpha$-parameters of the neurons. In Fig 8B we plot the magnitude of the eigenvalues of the reservoir matrix $W_{res}$. Both of these metrics govern the timescales at which the dynamics evolve. There is no clear difference between networks with and without the CPP. In Fig 8C we plot the ratio of excitatory

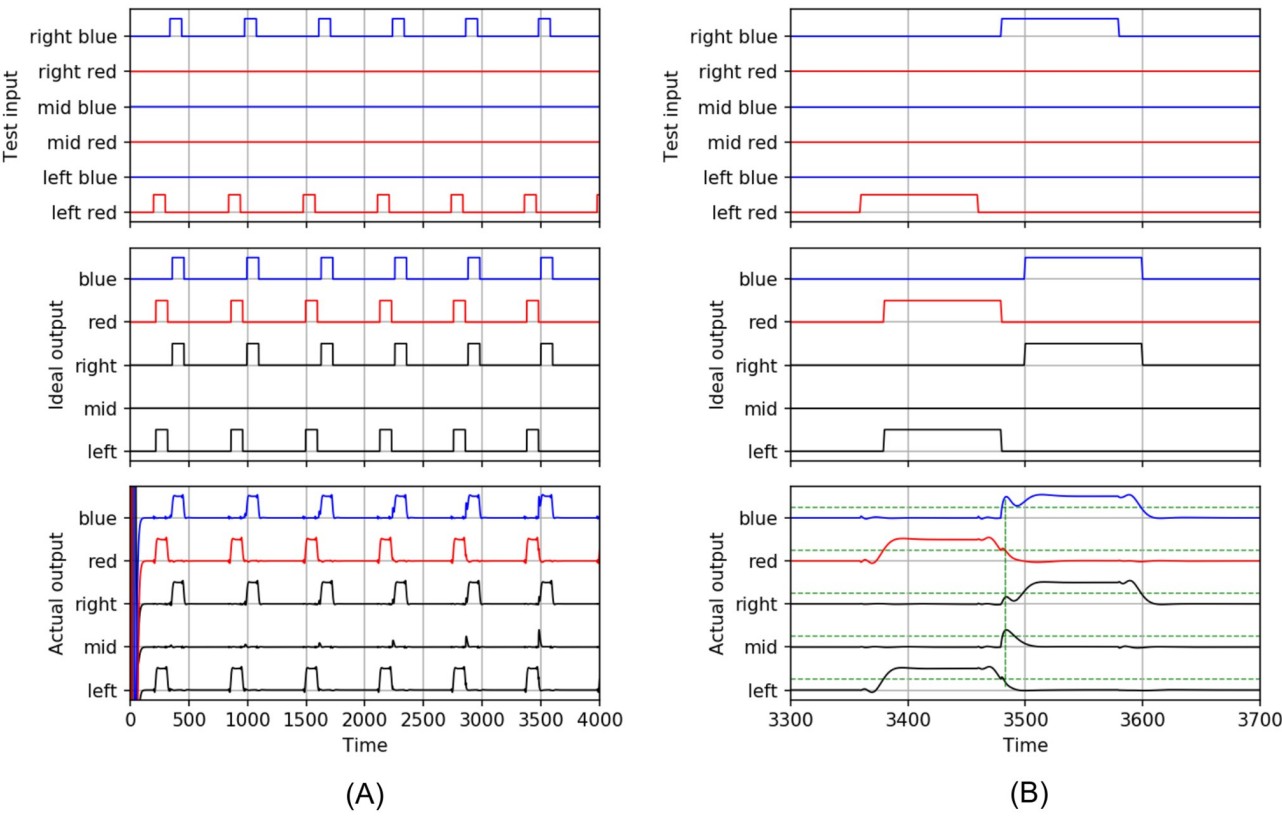

(A)

(B)

**Fig 7. Echo state network during testing for the color phi phenomenon.** (A) (top) Stimuli jumping between left-red and right-blue with decreasing intervals, (middle graph) ideal response with low middle output, (bottom graph) actual output showing CPP near $n \approx 3490$. (B) Zoom-in of (A). The vertical dashed green line in the bottom graph indicates the time at which the CPP occurs.

(positive) to inhibitory (negative) synapses. Finally, Fig 8D shows the number of positive long distance feedforward connections (LDFF) in the networks. Here, a connection from neuron $i$ to neuron $j$ is counted as LDFF, if $i$ positively influences $j$ via both a direct and an indirect path. This test was suggested by our phi model, where priming was caused by LDFF connections. Again, no differences between networks that show the CPP and those that do not come forward. These negative results suggest -but of course do not prove- that it truly is the detailed internal structure that matters, and the ESNs are otherwise macroscopically indistinguishable. We suspect only a small portion of the neurons in the ESNs assists in generating the CPP.

In Fig 9, we show a brief exploration of how robust the results are with respect to the choices of network parameters. Here, we scan over reservoir and input sparsity, spectral radius and input scaling, the latter being the $\sigma$-value with which the input matrix $W_{in}$ is constructed. The non-varied parameters are as described in the previous section. Each point is the fraction $p$ out of $n = 200$ ENSs that showed the CPP. The error bars are estimated at the 95% confidence interval using Wald's method: $p \pm z\sqrt{p(1-p)/n}$, with $z = 1.96$.

It must be remarked that invariably such parameters are highly interdependent, such that for the purpose of optimization they must be probed together. We did not explore this any further as this is was outside of the scope of this article.

In the next and final section we summarize our results and look at how they may be interpreted in light of the importance of the CPP.

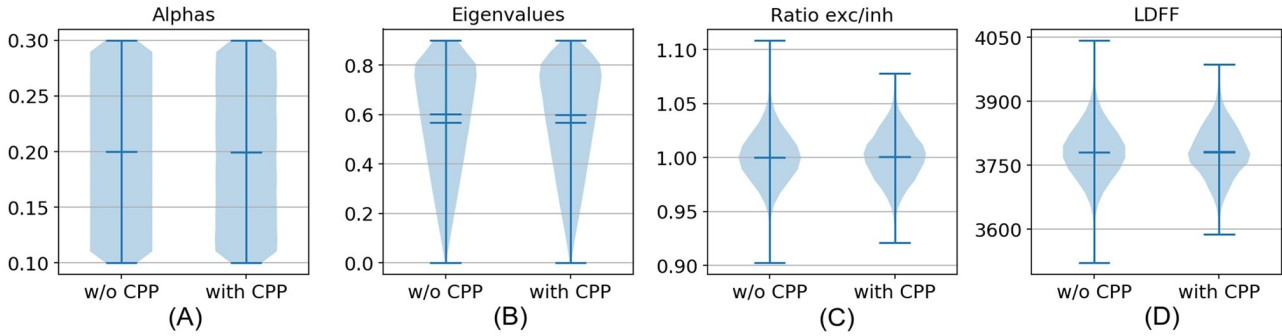

**Fig 8. Some characteristics of the ESNs that do ($N_{with}$ = 1867) and do not ($N_{without}$ = 98133) show the CPP.** (A) Distribution of neuron $\alpha$-timescales, (B) magnitude of the reservoir matrix eigenvalues, (C) ratio of excitatory to inhibitory synapses, (D) number of positive long distance feedforward connections.

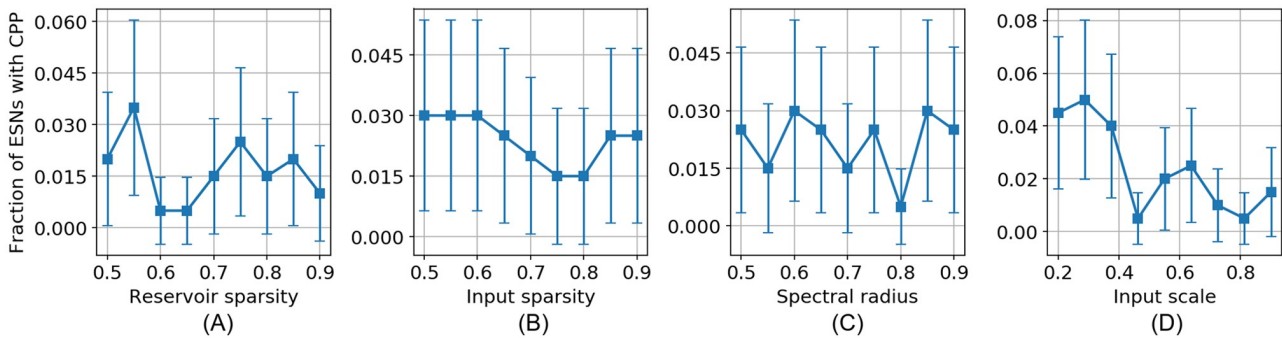

**Fig 9. Fraction of ESNs that show the CPP as a function of several network parameters.** (A) Reservoir sparsity, (B) input sparsity, (C) spectral radius, (D) input scaling. Each point represents $n$ = 200 tested networks. The error bars are estimated at the 95% confidence interval. Non-varied parameters kept as described in the text.

## Discussion

Our goal in this article was to show that small connectionist networks built from crude toy model neurons can capture the main aspects of the color phi phenomenon (CPP). The CPP was described by Dennett [19] as a landmark example of the constructive character of conscious experience: rather than merely recording events as they unfold over time, subjective experience involves both preconstructive and reconstructive mechanisms that constitute a coherent narrative for the subject whose experiences it is. There is thus no central "theater" in which unfolding events play out; rather, says Dennett, consciousness builds on "multiple drafts" out of which one narrative will eventually win out.

With this in mind, we simply asked which kinds of neural architectures might naturally lead to the striking reversals observed in the CPP. Our approach was based on the following assumption: if consciousness is something physical (as it must be), then there must be mechanisms through which objective stimuli are mapped into objectively measurable neuronal activations which either constitute or drive conscious perception in a way that is congruent with subjective reports. The core aspect of what needs to be explained in the case of the CPP is the chronological anomaly it presents us with: how can the color of the second stimulus be experienced before its position is? Here, we set out to explore which kinds of neural network architecture might produce such effects without further assumptions, and we showed that the

dynamical properties of relatively simple elementary networks naturally produce CPP-like effects. Our models are relatively simple circuits that are assembled from toy model neurons. They nevertheless have three essential properties in common with real biological neural systems: modularity, nonlinearity and time dependence. Our work is connected to the ideas recently developed by Herzog *et al.* [4], who proposed an explanation of visual perception that operates in two stages. The first stage consists of a continuous and unconscious integration of sensory information into features, best representing the stimulus. These features include time labels. During processing, all features are continuously updated yet not reportable. In a second stage, the features are rendered into consciousness all at once. We demonstrated stimulus-response order reversal in a four-neuron network that uses local recurrence, showing that, at least in this scenario, such time labels are unnecessary.

In line with Herzog *et al.*, our simple model shows continuous feature integration and a by definition discrete, threshold-like rendering of features into conscious percepts. Additionally, timing labels may be constructed as neuronal activaty, the value of which is related to the time since the onset of a (not yet reportable) stimulus. A thresholded signal derived from the stimulus as a whole acts as a gatekeeper. Naturally, a neural timing signal would have to be relative to the larger perceptual context or mental state, rather than present any objective wall clock time. In summary, the experience of time is just like any other percept: a construction. Our four-neuron time reversal model illustrates how little machinery is needed to falsify this experience. Thus we conclude that our models, if needed, could incorporate time label features as proposed by Herzog *et al.* [4].

Following the breakdown of possible types of temporal consciousness in Herzog *et al.* [33], we consider our models to come closest to the discrete retentional class, albeit with percepts of finite non-zero duration which is at least in part governed by the stimuli. In our ESN model, not all feature perceptions are updated simultaneously and this is where the CPP-like effect originates from: the "color" feature is updated (meaning; crosses a perceptual threshold) asynchronously from the "position" feature, see Fig 7B. Thus features can be updated all at once, without noticeable timing differences, but this is not a given, as it generally depends on the characteristics of the stimulus and probably also on the larger perceptual context. What our ESN model indicates is that the CPP is the result of dispersion between features that stimulus-wise belong to each other, but are processed by different pathways. Under the right conditions, subtle timing differences caused by the different velocities with which the features traverse the neural substrate cause a noticeable binding error.

Fekete *et al.* [34] have argued at length against a perception that is tied down in a "discrete straightjacket", on the basis of both empirical physiological data and philosophical arguments. One strong argument against a fully discrete awareness is the asynchronicity of inter area communications of the brain.

It is interesting to note that there are common elements in the structure of the order reversal and the phi phenomenon networks we presented. Both have local recurrences and long-distance feedforward connections. Local recurrences evidently may induce long lasting states without the need for large buffers and feedforward priming connections, as discussed in the section on the phi phenomenon model, can be interpreted as a form of prediction. Sadly, these simple models do not offer a clear path back into how they might be represented in the actual neural architecture of the brain, although functionally similar circuits must exist at some level.

We see several directions in which we could expand this work. First, it is clear that the interactions that take place in the networks we proposed are influenced by the amplitude, duration and interval characteristics of the stimuli, something we did not fully explore, except briefly for the introductory masking example and in Fig 3C. In the latter it was shown

that the relative strength of the stimuli influences the temporal perception of their intervals. Results from such explorations could be compared to empirical data with the goal of finding similar scaling laws.

Second, we did not offer any explanation as to how the proposed neuronal structures could have emerged in the first place. In other words: under which conditions might the relevant weights be learned instead of merely being hand-tuned? It is straightforward to equip the leaky integrator neurons with a learning rule that mimics spike-timing-dependent-plasticity, by setting $\dot{w}_{ij} = \eta x_i \dot{y}_j$, which can be interpreted as follows: when input $x_i$ is active while output $y_j$ increases, then this input *predicts* the increased output value, hence the associated weight should increase. This learning rule was recently proposed by Bengio *et al.* [35, 36]. It would be interesting to investigate under which circumstances the CPP emerges in a plastic neural network, as opposed to the ESN which has fixed randomly chosen internal weights.

In our models, we (naively?) interpreted the end of a percept as coinciding with crossing the same threshold as its onset but in the opposite direction. The animation S4 Animation in the supplementary information of how the ESN "perceives" the CPP, based on the timeseries of Fig 7, is constructed in this way. However, as many experiments have shown, there clearly is an asymmetry in the sensitivity of when something emerges vs. when it disappears from our perception, even in very simple visual scenarios. Such hysteresis effects have been hypothesized by Poltoratski and Tong [37] to disambiguate visual perception during the processing of scene transitions. In other words: the hysteretic thresholds are themselves a function of the scene in its entirety. Since a continuous signal -the precursor of a conscious percept- cannot pass two unequal values instantaneously, the existence of hysteresis naturally implies percepts of non-zero duration. Hysteresis effects could be explored further using the modeling approach shown in this article.

We stress again the crucial point that during the training phase of the ESN we explicitly excluded input conditions that could invoke a CPP: the readout layer was constructed from the network's responses to input stimuli that did not contain direct left-to-right or right-to-left jumps of opposite color. We found 1.87% of $N = 100000$ networks showed the CPP, with the network parameters we chose. Kim and Senjowski [38] showed that for a spiking recurrent neural network performing a working memory task, network characteristics that strongly influence the emerging timescales, and which are thus good predictors of performance, can be found. We similarly analyzed several characteristics of our ESNs, but failed to find markers that predicted the presence or absence of the CPP.

Although the ESN, or more general reservoir computing, has been described as a "brain-inspired" machine learning paradigm, one could argue that the readout layer, of which the construction is fundamentally based on a matrix inversion, is all but biologically plausible. Thus we can interpret the ESN simply as a function approximator which in itself has no relation to consciousness or the brain whatsoever. Stated otherwise: the ESN shown here exhibits sensitivity to input conditions that invoke the CPP, but remains mute concerning conscious perception. We conclude that the CPP, while an intriguing visual illusion in its own right, might be nothing more than a consequence of the inherent dynamics that characterize information processing in the brain.

## Supporting information

**S1 Animation. An animation showing the creation of stable states by feedback in a single toy model neuron network used for demonstrating priming.**
(WMV)

**S2 Animation. An animation of a bistable system with positive input stimulus masked by negative stimuli with decreasing interval.**
(WMV)

**S3 Animation. An animation of the phi phenomenon, the appearance of a "ghost" percept in dynamical neuronal network representing an abstract three pixel visual system.**
(WMV)

**S4 Animation. An animation of the color phi phenomenon based on the timeseries found in Fig 7A as "perceived" by the ESN during testing.**
(WMV)

**S1 Data. Contains the Python scripts that directly generate the graphs of this article.**
(ZIP)

## Acknowledgments

AC is a Research Director with the Fonds de la Recherche Scientifique F.R.S.-FNRS (Belgium).

## Author Contributions

**Conceptualization:** Lars Keuninckx, Axel Cleeremans.

**Formal analysis:** Lars Keuninckx.

**Funding acquisition:** Axel Cleeremans.

**Investigation:** Lars Keuninckx, Axel Cleeremans.

**Methodology:** Lars Keuninckx, Axel Cleeremans.

**Project administration:** Lars Keuninckx, Axel Cleeremans.

**Resources:** Lars Keuninckx, Axel Cleeremans.

**Software:** Lars Keuninckx.

**Supervision:** Axel Cleeremans.

**Validation:** Axel Cleeremans.

**Visualization:** Lars Keuninckx.

**Writing – original draft:** Lars Keuninckx, Axel Cleeremans.

**Writing – review & editing:** Lars Keuninckx, Axel Cleeremans.

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
