## [Decision Letter · Decision Letter 0]

27 Apr 2021

Dear Dr. Keuninckx,

First of all let me apologise once again for the delay. The current circumstances contribute to the further stretch of the queues of the turnover time distribution.

Thank you very much for submitting your manuscript "The color phi phenomenon: not so special, after all?" for consideration at PLOS Computational Biology.

As with all papers reviewed by the journal, your manuscript was reviewed by members of the editorial board and by several independent reviewers. The paper was overall very well received, but some issues need to be addressed. In light of the reviews (below this email), we would like to invite the resubmission of a significantly-revised version that takes into account the reviewers' comments.

We cannot make any decision about publication until we have seen the revised manuscript and your response to the reviewers' comments. Your revised manuscript is also likely to be sent to reviewers for further evaluation.

Sincerely,

Daniele Marinazzo

Deputy Editor

PLOS Computational Biology

Daniele Marinazzo

Deputy Editor

PLOS Computational Biology

Reviewer's Responses to Questions

**Comments to the Authors: **

Reviewer #1: Review is uploaded as an attachment.

Reviewer #2: The authors present an echo state model, which can capture the color phi phenomenon. The results are interpreted within the broader context of the philosophical questions about the time course of consciousness. The research questions is timely and interesting. The presentation, the specific research question, and the model in action were not always clear to me.

Majors.

It is not easy to get what the model exactly does because the presentation is sometimes quite wooly and at other, important instances very short. For example, why presenting a model that does not work? Why having this lengthy detailed description of integrate and fire neurons? Why not explaining the echo state network right away and more clearly. 

If I got it correctly, the model is more or less a low pass filter, which translates the input into a perceptual space producing some neural activity in the middle between, say the right blue and left red input, which mimics the phi phenomenon because there is activity at a location where nothing is produced. OK this is interesting.

However, what are the implications? It seems that the authors want to attack certain notions about the time course of consciousness by Dennett and us by showing that a simple model can do the job. It remains however unclear to me what exactly they want to attack. Or maybe they do not want to criticize things? At other instances it reads like that the authors want to naturalize philosophical thoughts. Please clarify. 

A simple motion detector can also detect apparent motion and even the phi phenomenon when equipped with a color detector. Thus, even more simple mechanisms may exist. What do we learn from the authors model then?

It seems to me also that the network does not perceive motion, it just creates a signal in the middle of the trajectory, isn't it? 

There seems to be also a misunderstanding about our labels. These are not Dennett's time stamps. Our model claims that motion and other temporal aspects of a stimulus are rendered conscious at one short moment as a vector of feature labels, such as (Shape=line, duration=100ms, motion=l-r, color=red). Hence, the motion is not perceived during the 100ms of the presentation, not even for 100ms, just as a number, namely, the output of a duration detector.

In this respect, it seems that the model is different from our model since it outputs a continuous stream. I have the feeling the authors want to defend a extensionalism (Dainton, Stanford Encycl.) and is a bit in the spirit of other models for example by Piper. Here, there could by a real controversy. These things are spelled out in Herzog et al. (2020).

I think also Dennett would agree with the model. His time stamps are for representation of time, a very different questions than the authors focus on.

In any case, I think the authors should make clearer what they are aiming for. 

Our model is mainly inspired by post-dictive effects as they occur in feature fusion, feature fusion with TMS, the sequential metacontrast paradigm (SQM), and other where a later presented stimulus integrates with a previously presented one. Hence, the question is to what extent the model can address such phenomena, for example, a simple feature fusion percept where a red disk followed by a green disk produces are yellow disk.

Minors:

"The core aspect of what needs to be explained in the case of the CPP is the subjectively experienced reversal of the order in which stimuli occur." Why reversal? There is no reversal in app motion. The first disk is perceived before the second one.

"We conclude that the CPP, while an intriguing visual illusion in its own right, might be nothing more than a consequence of the inherent dynamics that characterize information processing in the brain." The CPP is not an inherent dynamic. It may be explained by one. What is the argument here? That Dennett has claimed that it needs sophisticated processing? 

What do we learn from the subplot "neuronal state" in fig 6?

English is not always clear: For example: "When undriven by a stimulus ~x(t), the state \\falls" downward towards a local minimum of and scalar function V (~y) : R^K -> R, much like a golf ball rolling downward to the lowest point of a valley it happens to find itself in." Undriven = no stimulus presented. States don't fall. I think state is the wrong word here because at each time point of "falling" is a state. I think a reader who understands the Nabla Operator knows what an attractor state is and thus needs no metaphors. For those who do not know, I am not sure whether the metaphor helps. Mathematically: why is a global minimum reached and not a local one?

"Since it is not straightforward to manually construct a neuronal circuit that shows the complete CPP, we turn to neural network paradigms to find one." Rather void sentence. 

Lines 166 ff could be removed. This should be clear to readers of Plos Comp.

Figures are low resolution. Please present them in the text together with the captions. 

"merely artifacts of the inherent dynamical and nonlinear behavior of such systems." Why artifacts? Again, it seems that authors want to say "look we do not need complicated machinery" but is unclear who they think claimed otherwise. 

"In Ref. [7]," spell out please.

Mathis and Mozer, no year is given.

Signed

Michael Herzog

Dainton, B. (2018) Temporal consciousness. In The Stanford

Encyclopedia of Philosophy Winter 2018.

Michael H. Herzog, Leila Drissi-Daoudi, and Adrien Doerig (2020).

All in Good Time: Long-Lasting Postdictive Effects Reveal Discrete Perception. TICS. 

Piper, M.S. (2019) Neurodynamics of time consciousness: an

extensionalist explanation of apparent motion and the specious

present via reentrant oscillatory multiplexing. Conscious. Cogn.

Reviewer #3: The authors address what is on the face of it an intriguing mystery, the so-called color-phi illusion which has been argued to evidence postdictive processing or even discrete perception by some. Rather than offering yet another conceptual analysis, the authors take a much more sensible approach to the problem, and set out to show that this phenomenon could result from generic facets of the dynamics of artificial neural networks. The authors do well in presenting the rational of their (and similar) modeling approach, stressing the logic in first demonstrating the general nature of an effect in abstract modeling.

As a modeling device, the authors choose to utilize leaky integrator neurons, which are exhibit several of the properties one would want from neural building blocks not the least of which is a temporal aspect to their response. They proceed to present three simple models in increasing order of complexity, a model for temporal reversal, a model for the phi phenomenon, and a model incorporating both.

The authors modelling efforts and analysis are a timely addition to the recent discussion triggered by Herzog et al’s discrete model of perception. As such they expose a possible weakness in the original argument, namely that time reversal has to be construed as postdiction. However, I think the paper would benefit from some additional analyses as listed below before it is published:

Major comments:

I really like the later parts of the discussion. I think they highlight some major issues with the notion of imposing discreetness on neural dynamics, and the conceptual link between discreteness and consciousness. I recommend that this somewhat agnostic view be represented also in the earlier discussions.

Order reversal model: The authors say that the parameters had to be handpicked to achieve reversal – but why couldn’t they use simulation, and parametrization to produce a phase map to demonstrate the robustness of the effect? 

The authors acknowledge the limitations of their phi phenomenon model, but again do not directly assess the robustness of even this limited result (the movie is a step in the right direction).

Again in the CPP model we are not offered insight as to robustness – e.g. what is the effect of the sparseness parameter? The spectral radius? Etc. (i.e. do we still get CPP in 1.2% of random networks?)

Minor comments

“A four toy model neuron model” a four neuron toy model?

I’m not sure eq. 5 helps illustrating the point – it just complicates a very straight forward example.

Why does the output layer have to be conceived as a cartesian theater? Can’t it be taken to be as behavioral output, thus implying that sensory motor mapping is the substrate of representation?

Why is the ESN constrained not to respond to simultaneous stimuli?

As an aside - our recent paper offers some analysis and points to existing evidence from VSDI to the proposed mechanism for phi illusions (In the interest of saving time: a critique of discrete perception)

**Have the authors made all data and (if applicable) computational code underlying the findings in their manuscript fully available?**

Reviewer #2: Yes

Reviewer #3: None

PLOS authors have the option to publish the peer review history of their article (what does this mean?). If published, this will include your full peer review and any attached files.

Reviewer #1: **Yes: **Jake Gavenas

Reviewer #2: **Yes: **Michael Herzog

Reviewer #3: **Yes: **Tomer Fekete

**Have all data underlying the figures and results presented in the manuscript been provided?**

Reviewer #1: **No: **The article was a simulation study, so no data was collected. They say they will include the python code in a zip file in the supplementary material, but that is not done with this manuscript.
---

## [Decision Letter · Decision Letter 1]

13 Jul 2021

Dear Dr. Keuninckx,

Thank you very much for submitting your manuscript "The color phi phenomenon: not so special, after all?" for consideration at PLOS Computational Biology. As with all papers reviewed by the journal, your manuscript was reviewed by members of the editorial board and by several independent reviewers. The reviewers appreciated the attention to an important topic. Based on the reviews, we will accept this manuscript for publication.

The reason why this is still technically a "minor revision" decision is that we would like to give you the chance to address the residual comments by Dr. Herzog, which would likely bring to a clearer final product.

Sincerely,

Daniele Marinazzo

Deputy Editor

PLOS Computational Biology

Daniele Marinazzo

Deputy Editor

PLOS Computational Biology

[LINK]

Reviewer's Responses to Questions

**Comments to the Authors:**

Reviewer #2: The authors have clarified and improved many things. Still, I am not really happy with the presentation. The main problem is the imbalance of context, which makes it hard to get what is important. Basic, well known stuff is explained in great detail and other things, such as the Nabla operator (I guess most readers are not familiar with), are just mentioned en passant (the fix "readers familiar..." does not really work). I will not urge the authors to change it, if they do not want to. It is their paper.

The second thing it is still with the interpretation what the implications are. For example, in the abstract it is written that the "color phi phenomenon is...an artifact of inherent dynamics". This does not make to much sense too me. Why an artifact and not a feature? It seems the authors what to attack certain positions, but which ones? The term postdiction is indeed often used for mechanism rather than a term for psychophysical effects. What do the authors have in mind?

I have some small comments in the attached pdf.

Signed

Michael Herzog

Reviewer #3: The authors did a thorough job in addressing the reviewers comments, thus I gladly endorse it for publication.

**Have the authors made all data and (if applicable) computational code underlying the findings in their manuscript fully available?**

Reviewer #2: Yes

Reviewer #3: Yes

PLOS authors have the option to publish the peer review history of their article (what does this mean?). If published, this will include your full peer review and any attached files.

Reviewer #2: **Yes: **Michael Herzog

Reviewer #3: **Yes: **Tomer Fekete

Figure Files:

Data Requirements:

Reproducibility:

References:

---

## [Editor Report · Decision Letter 2]

12 Aug 2021

Dear Dr. Keuninckx,

We are pleased to inform you that your manuscript 'The color phi phenomenon: not so special, after all?' has been provisionally accepted for publication in PLOS Computational Biology.

Best regards,

Daniele Marinazzo

Deputy Editor

PLOS Computational Biology

Daniele Marinazzo

Deputy Editor

PLOS Computational Biology

---

## [Editor Report · Acceptance letter]

31 Aug 2021

PCOMPBIOL-D-21-00262R2 

The color phi phenomenon: not so special, after all?

Dear Dr Keuninckx,

I am pleased to inform you that your manuscript has been formally accepted for publication in PLOS Computational Biology. Your manuscript is now with our production department and you will be notified of the publication date in due course.

With kind regards,

Andrea Szabo
